# Biomarkers for Malignant Pleural Mesothelioma—A Novel View on Inflammation

**DOI:** 10.3390/cancers13040658

**Published:** 2021-02-06

**Authors:** Melanie Vogl, Anna Rosenmayr, Tomas Bohanes, Axel Scheed, Milos Brndiar, Elisabeth Stubenberger, Bahil Ghanim

**Affiliations:** Department of General and Thoracic Surgery, Karl Landsteiner University of Health Sciences, University Hospital Krems, 3500 Krems an der Donau, Austria; melanie.vogl@krems.lknoe.at (M.V.); arosenmayr@gmail.com (A.R.); bohanest@gmail.com (T.B.); axel.scheed@krems.lknoe.at (A.S.); milos.brndiar@krems.lknoe.at (M.B.); elisabeth.stubenberger@krems.lknoe.at (E.S.)

**Keywords:** malignant pleural mesothelioma, inflammation, infiltrating immune cells, prognostic biomarker, predictive biomarker, immune therapy

## Abstract

**Simple Summary:**

In view of the recent advances in immunoncology, we want to reevaluate and summarize the role of the immune system in malignant pleural mesothelioma (MPM). MPM is an aggressive disease with limited treatment options and devastating prognosis. Exposure to asbestos and chronic inflammation have long been acknowledged as main risk factors. In this review, we summarize the current knowledge about local and systemic inflammation promoting pathogenesis and progression of MPM. We focus on the prognostic and predictive value of infiltrating immune cells within the tumor and its microenvironment as local inflammation on the one hand and systemic inflammatory parameters on the other. We found that suppression of the specific and activation of the unspecific immune system are essential drivers of MPM, resulting in poor patient outcome. Numerous local and systemic inflammatory parameters are promising potential biomarkers for MPM, worth further research.

**Abstract:**

Malignant pleural mesothelioma (MPM) is an aggressive disease with limited treatment response and devastating prognosis. Exposure to asbestos and chronic inflammation are acknowledged as main risk factors. Since immune therapy evolved as a promising novel treatment modality, we want to reevaluate and summarize the role of the inflammatory system in MPM. This review focuses on local tumor associated inflammation on the one hand and systemic inflammatory markers, and their impact on MPM outcome, on the other hand. Identification of new biomarkers helps to select optimal patient tailored therapy, avoid ineffective treatment with its related side effects and consequently improves patient’s outcome in this rare disease. Additionally, a better understanding of the tumor promoting and tumor suppressing inflammatory processes, influencing MPM pathogenesis and progression, might also reveal possible new targets for MPM treatment. After reviewing the currently available literature and according to our own research, it is concluded that the suppression of the specific immune system and the activation of its innate counterpart are crucial drivers of MPM aggressiveness translating to poor patient outcome.

## 1. Introduction

Malignant pleural mesothelioma (MPM) is an aggressive neoplasm of mesothelial origin. Patients face a devastating prognosis of 12 months median survival only after diagnosis [1]. Despite recent—and, in part, promising—developments regarding both systemic therapy and cytoreductive surgery, MPM remains a clinical challenge, especially when it comes to treatment allocation [2,3]. Furthermore, the optimal (multimodal) treatment regimens still remain to be defined from the available arsenal of immune therapy, surgery, radiation and systemic treatment [4].

The pathogenesis of MPM was already associated with chronic inflammation, induced by asbestos exposure, sixty years ago by Wagner et al. [5]. Asbestos remains the main risk factor for developing this rare disease with a latency period of up to 40 years from time of exposure to diagnosis [1,6]. When inhaled, the long and thin asbestos fibers penetrate the lung parenchyma and deposit in the pleura, causing irritation and chronic inflammation. Consequently, the activation of surrounding immune cells leads to the secretion of cytokines, formation of reactive oxygen and nitrogen species, tumor necrosis factor α (TNF-α) release and nuclear factor ‘kappa-light-chain-enhancer’ of activated B-cells (NFκ-B) expression, in the end resulting in the accumulation of DNA damage and thus malignant evolution as reviewed before [6,7].

The activated immune system—especially with regard to its innate blood derived components—proved to be associated with worse patient outcome, late stage of disease, high Ki67 expression and poor treatment response in MPM as shown before by the authors and other research groups [8,9,10,11,12,13]. Not only for MPM but generally in oncology, the tumor promoting role of the immune system has been increasingly recognized as reflected in the latest version of the hallmarks of cancer by Hanahan and Weinberg [14]. Most recently, the immune system also evolved as a promising treatment target and modern immune therapy revealed as effective treatment modality in many solid tumors including MPM [15,16,17,18,19].

In light of the past and recent insights regarding the role of inflammation in the development and progression of MPM, inflammatory parameters are currently considered promising biomarkers [20]. In this review, we provide an overview about up to date knowledge of local inflammation in MPM and its involved immune cells as well as the tumor induced systemic inflammatory response. Special focus lies on the use of local and systemic inflammatory parameters as biomarkers for prognostic and predictive purposes in hope to facilitate and optimize treatment decisions and highlight new therapeutic targets for the future management of MPM. Predictive biomarkers might help to answer these crucial questions and are therefore desperately needed [21]. Despite the urgent need, to date there are no biomarkers recommended for MPM in daily practice in the current European guidelines since most studies failed to show sufficient reproducibility, sensitivity and specificity to justify the use of any suggested diagnostic biomarker so far. Unfortunately, the same holds true when it comes to prognostic, predictive or follow up biomarkers and thus further research is requested to better personalize treatment for MPM patients [22].

For this review we performed literature research in PubMed including English literature only. The following search terms were used: mesothelioma combined with prognostic and predictive biomarker, inflammation, inflammatory markers, C-reactive protein, fibrinogen, neutrophil to lymphocyte ratio, monocyte to lymphocyte ratio, thrombocyte to lymphocyte ratio, neutrophils, leukocytes, monocytes, albumin, Glasgow prognostic score, IL-6, ferritin, tumor microenvironment, tumor infiltrating lymphocytes, tumor associated macrophages/monocytes, PD-L1 and PD1, CTLA-4, immune therapy, and complement system. Since mesothelioma is a very rare disease and research regarding inflammatory biomarkers is limited, we included all available studies regarding biomarkers and only excluded case reports

Literature from the very early days of mesothelioma research ranging back to Wagner et al. from 1960 were included [5] as well as the most recent MPM literature from the beginning of 2021, resulting in 202 included references.

## 2. Findings

### 2.1. The Role of Local Inflammation in MPM

Several studies proved that (pre)malignant cells of various origins induce an inflammatory response with a paradox tumor promoting effect [23]. Local inflammation and immune cell infiltration within the tumor nests as well as the surrounding tumor microenvironment (TME) strongly influence the development and progression of numerous malignant diseases [23,24] including MPM as reviewed by Hendry et al. [25].

On the other hand, the immune system and its cellular components also play a protective role, especially with regard to acquired immunity as Leigh et al. observed already in 1986 correlating high lymphoid infiltration in mesothelioma specimens to a better prognosis [26]. In the past, the role of different infiltrating immune cells within MPM and the stroma has, therefore, become of increasing research interest since the immune system seems to be characterized here by a—not yet fully understood—duality [12]. Our adaptive immune system is protective against cancer development and spread [27], but it is also well documented that the immune system plays a crucial tumor promoting role in eventually all steps of malignant evolution by contributing to carcinogenesis, proliferation, angiogenesis, local infiltration and finally metastatic progression as reviewed by Coussens and Werb [28].

Very heterogenic immune cell infiltration in MPM tumor specimens and its TME has been described [29,30,31,32,33], with most studies reporting a predominant infiltration of tumor-associated macrophages (TAM) and tumor infiltrating lymphocytes (TIL), in particular CD4+ and CD8+ T-lymphocytes as reviewed by Chu et al. [34]. These cells are assumed to be the key players in the tumor associated immunoreaction. However, also rarer detectable myeloid derived suppressor cells (MDSC) [35,36], natural killer (NK) cells [28,31,32] and regulatory T cells (Treg) [29,36,37] have been studied before. These different immune cells infiltrating the tumor tissue but also contributing to the TME will be summarized in the following paragraphs as well as in Table 1 with regard to their role on MPM outcome and treatment response.

### 2.2. Tumor Infiltrating Lymphocytes (TIL)

TIL comprise T- and B-lymphocytes that have left the blood stream and infiltrated the tumor itself as well as the tumor stroma. Invading CD4+ T cells and proinflammatory cytokines prime CD8+ T cells to become effector cytotoxic T-lymphocytes (CTL), which then play a key role in eliminating cancer cells as reviewed before [48]. During tumor progression, cancer cells can avoid this effect by overexpression of programmed death ligand 1 (PD-L1) and cytotoxic T-lymphocyte antigen 4 (CTLA-4) (compare corresponding subchapter). Simultaneously, TIL release cytokines, thereby influencing various other immune cells, including the differentiation of TAM towards the immune suppressive type 2 macrophages (M2). This mechanism represents a negative feedback loop to avoid an over activated immune response. However, both aforementioned—in principal protective—immunosuppressive mechanisms (PD-L1/CLA-4 on the intercellular signaling level and type 2 macrophage differentiation on the cellular level) might lead to tumor immune evasion and thus uncontrolled tumor growth and progression [28,49,50].

For MPM, a predominant infiltration of CD8+ and CD4+ T-lymphocytes has been described by various researchers [30,31,32,51], but also the role of B lymphocytes [29] and Treg [51] is under investigation as described below.

The influence of B lymphocytes, key players in adaptive humoral immunity, is not fully understood and controversial results have been published so far for mesothelioma. Several studies reported low numbers of infiltrating B lymphocytes as reviewed by Minnema-Luiting [29]. Nevertheless, others discovered high CD 20+ B lymphocytes infiltration as well as the ratio CD163+ macrophages/CD20+ B lymphocytes as independent prognostic factors indicating better prognosis [37].

The prognostic value of CD8+ T lymphocytes, likewise part of the adaptive immunity, is better investigated and thus better understood for different MPM patient populations:

A number of studies investigated tumor samples of patients receiving trimodal therapy including cytoreductive surgery. Some reported an independent favorable prognostic value of high levels of CD8+ TIL [31,32], others found the ratio M2 count/CD8+ TIL count independently indicating negative prognosis [37], also suggesting that patients with low M2 and high CD8+ count have better outcomes. One other study found a correlation of local tumor overgrowth and low levels of CD8+ TIL in surgical patients [49] suggesting again an association with worse prognosis when the adapted immune system is underrepresented in the tumor compared to its innate counterpart.

On the contrary, Pasello et al. found high levels of CD8+ TIL in treatment naïve patients correlating not only with poor prognosis and aggressiveness of the tumor, but also a predictive value of high CD8+ TIL count for low response to chemotherapy. However, high levels of CD8 + TIL correlated additionally with high PD-L1 expression, which the authors speculate to be causal for the observed poor prognosis [38].

Additionally, high CD4+ T cell count in the tumor correlated with better outcome. Yamada et al. showed a tendency for better survival if CD4 + TIL and NK levels were high but did not reach level significance [32]. Marcq et al. compared treatment of naïve patients with those pretreated with chemotherapy and found high count of CD4+ TIL in the TME to be an independent positive prognostic marker for both therapy subgroups [30].

Treg, the immunosuppressive subset of CD4+ T cells, physiologically regulates immune tolerance, but also plays a major role in tumor development. Whereas only scarcely present in healthy tissue, a strong infiltration of Treg has been shown for many tumor entities [50] including MPM [51].

First data suggested that Treg and their deactivation via depletion of the surface marker CD25+ influenced survival in a murine model in a positive way [51]. Additionally, it was hypothesized that response to chemotherapy might be influenced by T effector cells and Treg [30].

While only a few studies analyzed the prognostic and predictive potential of Treg count in MPM, others investigated the cytokines responsible for Treg recruitment and activation, such as transforming growth factor (TGF-β) [6,52,53,54] and cyclooxygenase-2 (COX-2)/prostaglandin E2 (PGE2) [39,40,41,42,55,56], which are released by cancer cells directly or indirectly by cancer associated fibroblasts [50].

### 2.3. Cancer-Associated Fibroblasts (CAF)

CAF are abundantly present in numerous tumor entities and play a key role in the immunosuppressive effect of TME via cross-talk with Treg. High numbers of CAF are hence often associated with tumor promotion and poor prognosis. In turn, Treg stimulate resident fibroblasts to differentiate into CAF, which emphasizes the tight cross-talk between Tregs and CAFs [50].

A number of studies confirmed high numbers of CAF in TME from MPM samples [29]. As mentioned above, the major cytokines mediating CAF and Treg function have been under thorough investigation. Latest studies suggested a correlation between fibroblast growth factor (FGF) overexpression and high numbers of CAF with tumor aggressiveness and worse prognosis; however, the prognostic or predictive value is currently unknown and further research is obligatory [57,58]. Schelch et al. analyzed the role of different FGFs and their receptors in MPM in vitro and in vivo and proved that the FGF axis promotes cell proliferation, epithelial to mesenchymal transition, migration, invasion and clinical tumor aggressiveness. Inhibition of FGF receptor not only showed anti-proliferative effects itself but also a synergism with radiation and cisplatin and might, therefore, serve as a novel therapeutic target in MPM [58,59,60]. Furthermore, Li et al. also proved that FGF-2—besides platelet derived growth factor (PDGF) and hepatocyte growth factor (HGF)—is expressed by MPM. In addition, this study showed that MPM cell lines stimulate fibroblast motility and growth on the one hand and fibroblasts vice versa stimulate MPM growth and motility by HGF on the other hand, indicating an important cross-talk and tumor promoting symbiosis of CAFs and MPM cells [57].

### 2.4. Transforming Growth Factor-β (TGF-β)

TGF-β is known as an important inducer of CAFs and thus supporter of the immunosuppressive TME. Besides this tumor promoting characteristic, TGF-β can directly induce proliferation and epithelial to mesenchymal transition. In addition, TGF-β expression was associated with resistance to immune therapy as summarized recently [61]. With regard to MPM, TGF-β and its subtype activin A have been shown to be overexpressed in MPM cells with tumorigenic effects and thus inhibition or silencing was suggested as possible therapeutic target—first clinical results, however, were unsatisfactory with regard to fresolimumab, a TGF-β targeting antibody [6,53,54]. In addition, activin A blood levels were increased in MPM patients when compared to healthy controls and high activin A levels correlated with advanced tumor stage, high tumor volume and histological subtype translating to poor patient survival [54].

### 2.5. COX-2

Overexpression of COX-2 is detectable in various tumor cells and mostly associated with worse prognosis [62,63]. Nuvoli et al. reviewed the tumor promoting effects of proinflammatory prostaglandins, synthesized by COX-2 in general and for MPM in particular [55]. COX-2 overexpression was also found in MPM specimens [56]. Although some authors described controversial results regarding the prognostic value of COX-2, the majority reports a negative prognostic value of high tumor COX-2 expression [39,40,41,42].

In addition, the therapeutic effect of COX-2-inhibitors such as celecoxib has already been studied in other cancer types extensively [63]. However, COX-2 is not a routine target in modern oncology due to controversial results, e.g., for colorectal [64,65,66] and lung cancer [67,68]. For MPM on the contrary COX-2-inhibitors achieved promising results in vitro [35,57,69,70] and in vivo [36,71]. Unfortunately, neither NSAIDs nor COX-2 inhibitors prevented MPM development in an asbestos exposed risk group and in murine models [69]. The currently ongoing phase III randomized INFINITE trial assesses the effect of systemic celecoxib and chemotherapy combined with intrapleural INF-α (NCT03710876) and might answer the question whether COX-2 is an eligible treatment target in MPM.

### 2.6. M2 Macrophages

Under normal conditions, macrophages of the subtype M1 are part of the early inflammatory response enhancing the immune reaction while the immunosuppressive M2 macrophages limit a possible inflammatory overreaction [28,49,50]. A large proportion of M2 of total TAM consequently enforces tumor-promoting and immunosuppressive conditions and has been shown to indicate poor survival for different malignancies [70].

Additionally, for MPM specimens, various studies described strong infiltration of macrophages, predominantly of immunosuppressive M2, as prognostic marker [46,50,72]. High count of infiltrating M2 not only correlated significantly with poor prognosis [73] and increased proliferation rate but also reduced response to chemotherapy [74]. Others found no correlation between prognosis and absolute count of TAM or M2 but reported that high percentage of M2 of total TAM correlates significantly with local overgrowth [49] and negative prognosis [44].

In conclusion, current scarce data indicate that tumor infiltrating M2 might have prognostic and predictive potential. Interestingly, there is abundant research on M2 promoting cytokines and their impact on prognosis and treatment response.

Hematopoietic cytokines, including granulocyte macrophage colony stimulating factor (GM-CSF), were shown to be released by MPM cells especially when exposed to inflammatory cytokines but also autonomously [72,75] promoting the release of monocytes to the peripheral blood.

Additionally, cytokines promoting the M2 differentiation, namely IL-34 [45], macrophage colony stimulating factor (M-CSF) [74] and C-C motif chemokine ligand 2 (CCL2) [76,77] have been found to be elevated in tumor specimens or pleural effusion of MPM patients. High pleural levels of IL-34 correlated with worse prognosis [45], as well as high serum M-CSF, the latter also with response to chemotherapy [78]. Furthermore CCL2, a proinflammatory chemokine for monocyte recruitment, has been investigated over the past years. MPM patients showed significant higher serum levels of CCL2 than an asbestos exposed cohort without evidence of disease [79]. Similar results have been published by Gueugnon et al. as well as Blanquart et al. who found significant higher concentrations of CCL2 in pleural effusion of MPM patients compared to benign effusion or metastatic adenocarcinomas [76,77]. CCL2 released by MPM cells directly plays an important role in monocyte recruitment. CCL2 inhibition is also a potential treatment target and currently under investigation [50,78,80].

### 2.7. Myeloid-Derived Suppressor Cells (MDSCs)

The immune-suppressive MDSC are immature myeloid cells stimulated by tumor-derived cytokines. Abundantly detectable in MPM TME [35,36] they activate tumor-promoting Treg and inhibit tumor-suppressing CD4+ and CD8+ T cells [36]. A negative prognostic potential of MDSC can, therefore, be assumed; however, to our knowledge, no data regarding the prognostic or predictive value is currently available for MPM.

### 2.8. Natural Killer Cells and Dendritic Cells (DC)

The majority of studies reported low proportion of DC and NK in MPM specimens [28,31,32]. Yamada et al. additionally investigated the prognostic potential of NK infiltration and found no correlation with outcome [32]. Hegmans et al. confirmed a weak infiltration of DC, although they found a strong infiltration by NK. As possible explanation for low DC numbers they suggest the high levels of Interleukin-6 (IL-6) produced by MPM cells, since IL-6 inhibits the differentiation of progenitor cells to DC [51].

Summarizing these findings, DC and NK—both part of the innate immune system—are currently suspected to play a subordinate role in MPM and are, therefore, underrepresented in medical literature when compared to the aforementioned more prominent cellular players in the tumor and its TME.

### 2.9. Programmed Death Ligand 1 (PD-L1) and Cytotoxic T-Lymphocyte Antigen 4 (CTLA-4)

PD-L1 is expressed on the surface of various tumor cells and has the ability to bind to PD-1 receptors of CD4+ and CD8+ T Cells thus altering proliferation and cytokine production, leading to T cell inactivation and apoptosis of these important cellular players of adaptive immunity. As reviewed before by Zielinski et al. both PD-1/PD-L1 and CTLA-4 act as similar pathways downregulating lymphocyte response and accordingly adaptive immunity [80]. This tumor immune evasion results in progression and poor prognosis of various solid tumors [81]. With regard to thoracic oncology, the PD-L1 axis and its prognostic role were already analyzed in malignant pleural effusion [82] and stage IV lung cancer [83,84]. Furthermore, PD-L1 expression showed also prognostic potential in MPM as summarized in a recent meta-analysis [85].

PD-L1 positivity in MPM tumor cells was reported at heterogeneous levels ranging from 16 to 68% [29,47,86]. According to Marcq et al., PD-L1 and PD-1 are decreased after chemotherapy [30]. However, for peritoneal mesothelioma, controversial effects of chemotherapy on PD-L1 expression were reported [87]. PD-1 expression on TIL was furthermore described as a negative predictive factor for response to chemotherapy [30]. Additionally, PD-L1 was found to correlate with the sarcomatoid and biphasic histology of MPM [86]. Our study group recently showed that tumor PD-L1 expression is not only prognostic in an international cohort suffering from malignant pleural effusion (in part also caused by MPM) but was significantly interacting with CRP, thus suggesting that the prognostic values of both markers influence each other. This observation translated to the poorest survival in the patient group characterized by high CRP in the patient blood and high PD-L1 expression in the corresponding tumor specimen [82]. Inaguma et al. demonstrated the independent prognostic impact of PD-L1 and activated leukocyte cell adhesion molecule (ALCAM, CD166) in MPM. Expression of both led to the shortest overall survival (OS). Additionally, a significant association between PD-L1 and ALCAM was drawn [46]. Similar prognostic results have been shown previously [43,47].

To reverse the tumor promoting effect of a downregulated specific immune system, checkpoint inhibitors like humanized monoclonal antibodies against PD-1 or PD-L1 have been developed. Immune evasion can be stopped to increase tumor defense [16,81]. The therapeutic benefit of targeting PD-1 with pembrolizumab [16] or nivolumab [17], and PD-L1 with avelumab [18], in pre-treated MPM patients with PD-L1 positive tumors was demonstrated.

The aforementioned other—by malignant disease misused—pathway of adapted immunity downregulation, namely CLTA-4 has also been investigated and proved to be an interesting treatment target to reactivate the immune system against MPM progression as reviewed before [88]. More recently, the combination of nivolumab with ipilimumab was approved by the FDA for unresectable MPM as first line therapy according to promising results documented during the CHECKMATE 743 trial, indicating that the combination of PD-1 and CLTA-4 targeting immune therapy is effective in MPM [19].

Finally, soluble PD-L1 (sPD-L1) from the sera of patients before and during immune therapy was suggested as a predictive biomarker, indicating poor treatment response when elevated before and during immune therapy. Additionally, sPD-L1 levels were also correlating to the inflammatory parameters NLR, neutrophil count and CRP, blood parameters that will be described later on in more detail [15]. Most recently, the role of sPD-L1 was also investigated in pleural effusions [89]. Both serum as well as pleural effusion derived PD-L1 status might represent an easily available method for clinical monitoring of the treatment target during immune therapy in the future.

Although there is great hope for a more personalized immune therapy, the exact background of the heterogeneity in PD-L1 expression and in treatment response is not yet fully understood. The interplay of tumor immunology, immunotherapy and somatic mutations is currently intensively researched [87,90,91,92]. Yang et al. recently reviewed the complex interactions of molecular characteristics of MPM cells and TME with histological subtype and genomic mutations [93] underlining the need for a deeper understanding of the pathobiological processes in MPM in order to optimize personalized biomarker-guided immunotherapy.

The complex interactions of tumor infiltrating immune cells with MPM cells as well as the resulting systemic inflammatory processes—which will be discussed in the next chapter—are also graphically shown in Figure 1.

In summary, the role of local inflammation and the components of TME in MPM have been investigated by various researchers. We encountered promising data regarding the prognostic potential of the different tumor infiltrating immune cells and also first results for predictive potential of some of these biomarkers. However, most interestingly, we noticed that generally low numbers of specific immune cells as well as high numbers of unspecific immune cells seem to be unfavorable, suggesting a controversial impact of the innate and adaptive immune cells on local tumor progression.

### 2.10. The Role of Systemic Inflammatory Response in MPM

Systemic inflammation is becoming an increasingly acknowledged factor in the development and progression of different solid tumors, including MPM. Consequently, peripheral blood derived inflammatory markers, which are determined routinely in daily practice for almost all patients, have been extensively examined regarding their applicability as biomarkers in MPM as reviewed before [12].

Since systemic inflammatory parameters can indicate inflammatory and infectious processes in the patient’s body as well as malignancy, they are highly unspecific for diagnostic or screening purposes. However, after exclusion of acute inflammation or infection, some of the established and widely available inflammatory markers have been identified as prognostic or predictive markers in various solid tumors [94,95,96,97,98,99] including MPM [100,101,102].

As mentioned before, the current European guidelines for MPM management do not recommend any prognostic biomarkers for clinical use [4]. Nevertheless, two prognostic scores have been developed that are widely accepted and well established, namely the EORTC score (European Organization for Research and Treatment of Cancer) [103] and the CALGB score (Cancer and Leukemia Group B) [104]. Both scores have been validated for MPM by different researchers and proved their reproducibility [105,106,107,108]. Interestingly, these two scores not only integrate clinical, pathological and epidemiological factors, but also acknowledge systemic inflammation as tumor aggressiveness criteria by including the blood characteristics leukocytosis, thrombocytosis and elevated C-reactive protein (CRP) as negative prognostic factors [103,104]. The following paragraphs discuss the current literature on systemic inflammatory markers in MPM as also summarized in Table 2.

Leukocytosis, a well-known biomarker of acute inflammation, has been widely investigated as biomarker for MPM and a number of studies reported a negative prognostic value of elevated pretreatment white blood cell count after uni- [103,111,112,116] and multivariate [102,110,115] survival analyses. Absolute lymphocyte count, as sign of an activated specific immune system, was studied by fewer researchers as single biomarker, but an association with poor prognosis and reduced clinical response to chemotherapy has been reported so far [109]. However, the role of lymphocyte count on MPM outcome has been investigated more intensively with regard to different ratios, especially the neutrophil to lymphocyte ratio (NLR) which will be later described more in detail.

Monocyte count on the contrary has been studied more extensively as single prognostic marker in MPM. Burt et al. found an independent negative prognostic value of pretreatment monocytosis for patients undergoing cytoreductive surgery [73] and Zhang et al. and Tanrikulu et al. confirmed these findings for patients receiving different therapies [100,110]. Monocytes, as part of the unspecific immune system, are the procurer cells of tissue specific macrophages [73] including TAM who play an important role in the TME and thus contribute to local immune modulation as mentioned before.

Interestingly, neutrophil count, likewise representing the unspecific immune response, is rarely reported as single blood marker. Few studies describe controversial results, reporting adverse prognostic value of high neutrophil count in univariate analysis [100] or no correlation with prognosis [109,110]. Nevertheless, the neutrophil count compared to the lymphocyte count is more intensively studied when it comes to the NLR.

Thrombocytosis, a known unspecific systemic phenomenon in response to inflammation [126], has long been suggested as independent prognostic factor. Already in 1989, first data suggested an independent negative prognostic value of high platelet count [111], which has been confirmed by others in the following decades [109,110,111,116,117]. Other studies could not validate the prognostic value at all [112], or found, instead of platelet count, the platelet to lymphocyte ratio (PLR) to be prognostic, as explained more in detail below [102,103,104,105,106,107,108,109,110,111,112,113,114].

Next to single blood parameters, special focus has lately been laid on ratios between some blood markers, such as the neutrophil to lymphocyte ratio (NLR), lymphocyte to monocyte ratio (LMR) or platelet to lymphocyte ratio (PLR). These markers are easily accessible and calculated from routine blood cell count and reflect the relation between specific and unspecific systemic immune response.

With increasing knowledge of the role of specific and unspecific immune response in cancer, these ratios have become of rising interest as possible biomarkers in numerous malignancies with promising prognostic potential [94,113,114,115,116,117,118,119].

As for other solid tumors [94,113,114,115,116,117,118,119,120], a negative prognostic value of high NLR has been shown for MPM in numerous studies analyzing cohorts of patients receiving different therapy concepts [102,103,112,119,121] including systemic therapy [121]. Furthermore, two studies found in a subgroup analysis that normalization of pretreatment elevated NLR under chemotherapy was predictive for better OS [106,121]. Additionally, for surgical patients undergoing cytoreductive surgery, high NLR was found to correlate with worse prognosis [122].

Low LMR, displaying a domination of unspecific monocytes, has been found to be a negative prognostic marker for numerous malignancies as reviewed by Gu et al. [123]. For MPM, comparable results have been published [102,114,122] showing that low LMR is associated with adverse prognosis in line with the reported negative prognostic value of elevated monocyte count as mentioned before. Of note, Yin et al. published comparable results for peritoneal mesothelioma [124].

Furthermore, high PLR has been studied and reported as a negative prognostic marker for multiple malignancies [119,125,127,128,129], also including MPM. As already indicated above, of the four named studies with no correlation between platelet count and survival, three, however, did find PLR to be associated with poor prognosis after univariate analyses [102,103,114]. Thus, one might speculate that even if absolute platelet count alone is not prognostic, a relative increase in platelets compared to low lymphocytes might be.

### 2.11. Acute-Phase Proteins

Already in 1998, Nakano et al. observed significantly elevated serum levels of some acute phase response proteins (APP) and cytokines—namely fibrinogen, IL-6, alpha1-acid glycoprotein and CRP levels—in MPM patients compared to patients with adenocarcinoma of the lung. They also reported significantly higher levels of IL-6 in the pleural fluid of MPM patients and concluded that the pleural IL-6, when entering systemic circulation, enhances the systemic acute phase response (APR) [130].

The APR, as part of the unspecific immune response, is the physiological and biochemical systemic reaction to inflammation, infection, tissue damage due to, for example burn injuries or trauma and malignancies. The process is mediated by proinflammatory cytokines, causing fever, leukocytosis and the release of APP. Gabay et al. provide a detailed list of well-known APP, some of which have been under investigation with regard to applicability as inflammatory biomarkers in MPM—particularly IL-6, CRP, fibrinogen [126].

### 2.12. Interleukin 6

The proinflammatory cytokine IL-6 is released by various immune cells triggered by IL-1β and TNF, but also produced by tumor cells directly as also proven for MPM with tumor-promoting effects [75,131]. The (patho)physiological functions of IL-6 are reviewed by Hunter in general [132] and by Abdul Rahim et al. for mesothelioma in particular—emphasizing the promoting effect of IL-6 on cell proliferation, angiogenesis via stimulation of VEGF expression, resistance to chemotherapy and physical symptoms negatively influencing wellbeing of the patient [133].

In contrast to other malignancies [134,135,136,137,138,139,140,141,142,143,144] current data does not support the prognostic or predictive value of IL-6 serum concentration for MPM [130]. However, IL-6 levels have been reported to correlate significantly with other markers [130,133] of verified prognostic impact for MPM such as VEGF [145,146,147], thrombocytosis [107,111,142] and CRP levels [11,116,142]. Adachi et al. found that IL-6 encouraged cell proliferation as autocrine growth factor and the expression of VEGF [148] and accordingly investigated an IL-6 inhibitor as VEGF targeting therapeutic approach in a subsequent study [149].

Antiangiogenic therapeutic approaches have been widely investigated as reviewed recently by Novak et al. [150]. Thus, the clinical use of the VEGF inhibitor bevacizumab is now also regarded as promising improvement of the almost 20 year old standard chemotherapy regimen published by Vogelzang et al. [151] according to the promising results of the MAPS trial [152].

From the current point of view, IL-6 does not seem to be applicable as prognostic or predictive marker for MPM; however, it can be assumed that it plays a major role in promotion of systemic inflammation with release of other proinflammatory cytokines as already suggested two decades ago by Nakano et al. [130].

### 2.13. C-Reactive Protein (CRP)

CRP, first described in 1930, is one of the earliest discovered and most established acute-phase response proteins [153]. The CRP synthesis in hepatocytes is mainly stimulated by IL-6, IL-1β and tumor necrosis factor α (TNF-α) [126]. Clinical use for inflammation and treatment response is currently well established since elevated CRP levels correlate with the course of chronic and acute infections but also inflammatory (autoimmune) disorders, general tissue injury and various malignancies [154,155]. Lately, elevated serum CRP levels were found to be associated with poor prognosis for multiple tumor entities [82,94,95,97,156,157,158]. Consequently, this potentially interesting biomarker has also been investigated in MPM. Elevated CRP levels were reported to be associated with shorter survival—regardless of different applied treatment modalities [103,116,123], specifically for patients receiving systemic treatment [112], as well as patients undergoing trimodal therapy including surgery [11]. Some groups even described a level dependent negative prognostic potential of pretreatment CRP serum concentration [112,159].

Furthermore, the predictive potential of CRP for MPM has been explored by the study group of the authors before. It was proven that of patients undergoing aggressive multimodality treatment including cytoreductive surgery only those with normal pretreatment CRP levels benefit from this type of therapy. Thus, patients with normal CRP values before therapy receiving multimodality therapy survived 36 months in median. In contrast to these findings, patients with elevated pretreatment CRP only had 10 months overall survival despite multimodality therapy indicating, that indeed this subgroup of MPM is of distinct treatment responsiveness [11]. Of note, Kao et al. additionally described a correlation between elevated inflammatory markers—specifically elevated CRP and NLR—and advanced clinical symptoms such as fatigue and anorexia in course of an engraved systemic inflammatory response and consequently compromised health-related quality of life [160].

### 2.14. Fibrinogen

Fibrinogen, a well-known clotting factor, is also an important positive acute phase protein. Its synthesis is increasing significantly when stimulated by proinflammatory cytokines, mainly IL-6 [161]. Fibrinogen as biomarker has been investigated for several diseases including chronic obstructive pulmonary disease and coronary heart disease [161,162]. Additionally, a negative prognostic value of high pretreatment fibrinogen has been found for numerous tumor entities [98,114,163,164,165,166,167,168]. So far, only the previous study of the authors reported not only a prognostic but also predictive value for pretreatment fibrinogen in MPM. Patients with high fibrinogen plasma levels were shown to have significantly shorter OS. Additionally, of patients receiving trimodal treatment with cytoreductive surgery, those with high pretreatment fibrinogen did not benefit from multimodality treatment [10].

### 2.15. Albumin—A Negative Acute Phase Protein

Serum albumin not only reflects nutritional status but also inflammatory response as negative acute phase protein mediated by cytokines including IL-6, IL-1β and TNF-α [169]. Hypoalbuminemia has long been acknowledged to impair wound healing and outcome after interventions and surgeries [170,171,172,173,174]. In addition, it was described to indicate short survival in different malignancies [169]. For MPM, hypoalbuminemia has been associated with poor survival for patients receiving different treatment modalities [102,103,114], but also selectively for chemotherapy patients [175] and surgical patients [176].

In a classification and regression tree analysis, Brims et al. found the best survival for patients with no weight loss, a high hemoglobin level and a high serum albumin level [177]. Harris et al. validated these findings for surgical patients undergoing cytoreductive surgery [176].

Hypoalbuminemia and elevated CRP concentration have been integrated in a systemic inflammation based prognostic score, the so-called modified Glasgow Prognostic Score (mGPS). Its prognostic value has also been confirmed for multiple cancer types as reviewed in detail by McMillan [178] and has been acknowledged for mesothelioma in univariate analysis [101].

Furthermore, the prognostic value of elevated CRP/Albumin ratio (CAR), reflecting increased CRP values and decreased albumin concentration as indicator of poor nutritional and activated acute phase response, has been widely explored. Elevated CAR has been shown to predict poor outcome in acute diseases including sepsis [179,180] as well as in various malignant diseases [96,99,181,182,183,184]. Takamori et al. investigated CAR for MPM patients and found a high CAR to be independently prognostic [185]. Otoshi et al. confirmed these results for inoperable MPM patients [102] whereas Tanrikulu et al. could not reproduce these results [100].

### 2.16. Ferritin

The positive APP ferritin is up-regulated under inflammatory or infectious conditions to reduce the iron accessibility of pathogenic organisms [186,187]. For numerous malignancies elevated serum ferritin concentrations have been reported as well, in part with prognostic impact [188]. Healthy human cells, foreign organisms but also cancer cells depend on iron supply for a number of cellular metabolic processes. The role of iron metabolism and its regulation—partially by cells of the TME—have been reviewed excellently by Hsu et al. for cancer in general [188] and by Toyokuni et al. for MPM in particular, especially in context of asbestos-induced oxidative stress [189]. MPM has been associated with elevated ferritin serum levels [190,191,192], but to our knowledge the prognostic or predictive impact of ferritin has not been investigated so far. However, correlations of ferritin with TAM and modulated lymphocyte function has been suggested [187,188] so in context of the APR as well as the importance of iron metabolism in MPM the study of ferritin as biomarker might reveal interesting new results. Of note, also reduction of iron storage was suggested as possible treatment target after promising preclinical results from a rat model [192,193].

### 2.17. The Complement System

With regard to the innate immune system and its systemic circulating compartments, the complement component 4d (C4d) was also found to be of prognostic relevance in MPM patients. High plasma C4d levels were associated with high tumor volume, worse response to induction therapy, high acute phase response proteins and shorter survival after multivariate analyses as reported by Klikovits et al. [8]. Furthermore, Agnostinis et al. investigated the role of complement protein C1q in MPM. It was shown that C1q did not activate the classic complement pathway in MPM as one might expect, but instead bound to hyaluronic acid and thereby induced cell adhesion and proliferation of mesothelioma cells. Interestingly, the activation of the classic complement pathway was abandoned by hyaluronic acid [194]. These findings are in line with Klikovits et al., where high C4d (as downstream target of C1q during the classic complement pathway) was not correlating with high C1q [8]. Thus, the activation and exact role of the complement system and its subunits is yet not fully understood and might be of future interest in MPM.

Taken together, many common systemic inflammatory parameters have been studied regarding their prognostic potential for MPM and some additionally for their predictive impact. It is remarkable—compatible with our conclusions on local inflammation—that high unspecific inflammatory markers seem to be adverse whereas high specific inflammatory markers appear to be beneficial reflecting the tumor-promoting influence of the innate immune system and the tumor-suppressing impact of the adaptive immune system, respectively.

## 3. Conclusions

While preparing the present review and summarizing the established as well as most recent knowledge, it became fairly clear that a large amount of research considering this topic has been performed within the past few decades. Despite the fact that a lot of data is based on retrospective studies—which is most likely explainable by the rare incidence of MPM—high quality research supports the important role of inflammation in MPM. Not only in the setting of pathogenesis, tumor promotion, poor prognosis or treatment response inflammatory processes play a decisive role but inflammation and immune response are also under investigation as promising treatment targets. Most markers and key findings were not only published once but have been validated in the past, thus resulting in several inflammatory related biomarkers characterized by reproducibility and accordingly reliability. Furthermore, comparable results have been documented in other malignancies thus indicating, that some of the above mentioned findings have a universal impact in (immune-) oncology.

In the clinical management of MPM, physicians are confronted with multiple—yet not fully standardized—treatment modalities on the one hand, opposed to poor outcome and treatment resistance on the other hand. Consequently, MPM in general is the ideal candidate for biomarker research especially when it comes to treatment guiding predictive parameters [22].

The immune system plays a key role in MPM, since this rare disease has already been associated with inflammation several decades ago [5]. This theory is supported by many more recent studies summarized in this review. During the past few decades and especially within the last years, it was shown that an upregulated unspecific immune response on the one hand translates to poor outcome. On the one hand, a downregulated specific immune system results in tumor progression, tumor immune evasion and finally poor prognosis, regardless if investigating the inflammatory status in patient blood, pleural effusion, tumor tissue or its associated TME. Thus, the tumor suppressive characteristics of the specific immune system get obvious when the—through MPM suppressed—specific immune response gets reactivated by immune therapy resulting in prolonged survival.

This above-described duality of the immune system in MPM has been analyzed and described before by Linton et al. [12]. However, today we would reply to the question “Inflammation in malignant mesothelioma—friend or foe?” that it is both friend and foe. More precisely, and to simplify the key message from this review, the specific immune system is our friend and its unspecific counterpart the foe which is also reflected in the prognostic value of the corresponding biomarkers—both on a local as well as systemic level.

Further research on the immune system in MPM might help in treating this therapy refractory disease and reveal modern insights in the complex interaction of our immune system with the tumor thus resulting in a better biological understanding, new treatment approaches and finally improved clinical management and patient outcome. There is still need for future studies to gain detailed knowledge about this topic and, thus, we look forward to learning more about the interaction of our immune system with malignant disease in general and MPM in particular.

## Figures and Tables

**Figure 1 cancers-13-00658-f001:**
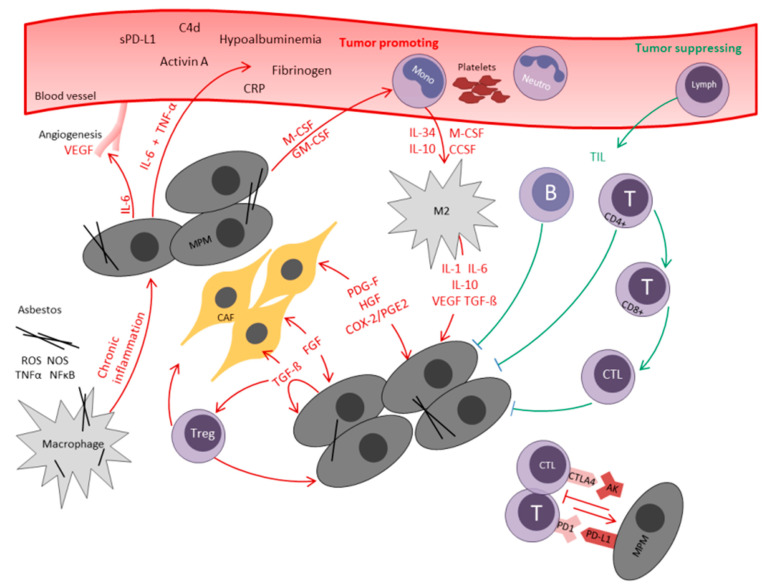
Interaction of local and systemic immune response in malignant pleural mesothelioma. *AB* antibody*, B* B-lymphocyte, *C4d* circulating complement component 4d, *CAF* cancer associated fibroblast, *CCSF* C-C motif chemokine ligand 2, *COX-2* cyclooxygenase-2, *CRP* C-reactive protein, *CTL* cytotoxic T-lymphocyte*, CTLA4* cytotoxic T-Lymphocyte Antigen 4, *FGF* fibroblast growth factor, *GM-CSF* granulocyte macrophage colony stimulating factor, *HGF* hepatocyte growth factor, *IL* interleukin, *Lymph* lymphocyte, *M2* M2-macrophage, *M-CSF* macrophage colony stimulating factor, *Mono* monocyte, *MPM* malignant pleural mesothelioma, NFκB nuclear factor kappa-light-chain-enhancer B, *Neutro* neutrocyte, *PD-1* programmed cell death protein 1, *PD-L1* programmed cell death ligand 1, *PDGF* platelet derived growth factor, *PGE2* prostaglandin E2, RNS reactive nitrogen species, *ROS* reactive oxygen species, *sPD-L1* soluble programmed cell death ligand 1, *T* T-lymphocyte, *TGF-ß* transforming growth factor-ß, *TIL* tumor-infiltrating lymphocyte, *TNF-α* tumor necrosis factor α, *Treg* regulatory T-cell, *VEGF* vascular endothelial growth factor.

**Table 1 cancers-13-00658-t001:** Potential local inflammatory biomarkers.

Biomarker	Unfavorable	Univariate Value	Multivariate Value	Impact	Design	Number of Patients/Range	References
B-TIL	Low	HR: N.R.	HR: 0.7	Prog	R	217	[37]
CD8+ TIL	Low	HR: N.S.-N.R.	HR: N.S.-0.4	All prog	All R	16–32	[30,31,38,39]
CD8+ TIL	High	HR: N.R.	HR: N.R.	Prog	R	93	[38]
M2/CD8+ TIL	high	HR: N.R.	HR: 1.6	Prog	R	210	[37]
M2/B-TIL	Low	HR: N.R.	HR: 1.6	Prog	R	210	[37]
CD4+ TIL	Low	HR: N.S.-N.R.	HR: N.S.-N.R.	All prog	All R	27–218	[30,31,32,37,38]
COX-2	High	HR: N.R.-2.9	HR: N.S.-4.6	All prog	R/R/P	29–77	[39,40,41]
COX-2	Low	HR: N.R.	HR: N.R.	Prog	R	86	[42]
M2	High	HR: N.S.-1.7	HR: N.S.	All prog	All R	4–210	[38,39,43]
M2/TAM	High	HR: N.R.	HR: N.S.	Prog	R	8	[44]
IL-34	High	HR: N.R.	HR: N.R.	Prog	R	74	[45] *
M-CSF	High	HR: N.R.	HR: N.S.	Prog	R	74	[45] *
PD-L1	High	HR: N.S.-N.R	HR: N.S.-2.3	All prog	All R	33–106	[17,30,43,46,47]

TIL tumor infiltrating lymphocyte, M2 macrophage subtype 2, Treg regulatory T cell, FGF fibroblast growth factor, TGF-β transforming growth factor β, COX-2 cyclooxygenase 2, TAM tumor associated macrophages, IL-34 interleukin 34, M-CSF macrophage colony stimulating factor, NK cells natural killer cells, PD-L1 Programmed cell death ligand 1, HR hazard ratio, N.R. not reported, N.S. not significant, Prog prognostic biomarker, Pred predictive biomarker, R retrospective, P prospective.* measured in pleural effusion.

**Table 2 cancers-13-00658-t002:** Potential Systemic Inflammatory Biomarkers.

Biomarker	Unfavorable	Univariate Value	Multivariate Value	Impact	Design	Number of Patients/Range	Cut-Off Value	References	
WBC count	High	HR: N.S.-1.9	HR: N.S.-2.3	All prog	All R	84–363	8.1–15.6 10^9^/L/8.3 10^9^/L *	[102,103,107,109,110,111,112,113,114,115,116,117]	
Lymphocytes	Low	N.S.-N.R.	N.S.	All prog	All R	105–285	1.27–2.0 10^9^/L	[102,114,117]	
Monocyte count	High	HR: N.R.-4.0	HR: N.S.-2.7	All prog	All R	105–667	0.55 10^9^/L	[43,102,114]	
M-CSF	High	HR: 1.6	HR: N.S	Prog	R	36	1120	[79]	
Neutrophil count	High	HR: N.S.-N.R.	HR: N.S.	All prog	All R	105–285	5.3–5.89 10^9^/L	[102,114,117]	
Platelet count	High	HR: N.S-2.1	HR: N.S.-2.1	All prog	All R	84–363	300–450 g/L, 400 10^9^/L *	[102,103,107,109,110,111,112,113,114,115,116,117,118]	
NLR	High	HR: N.S.-2.3	HR: N.S.-2.7	All prog	All R	30–285	3 and 5/5 *	[102,103,104,109,110,111,112,113,119,120,121]	
NLR normalization after treatment	No	HR: N.R.-2.2		All prog	All R	66–69	Decline to 5	[109,111]	
LMR	Low	HR: N.R.	HR: N.S.-1.8	All prog	All R	105–283	2.36–2.74	[102,114,122]	
PLR	High	HR: N.R.-1.5	HR: N.S.	All prog	All R	105–285	144–300	[102,103,114]	
CRP	High	HR: N.S-2.8	HR: N.S.-2.7	All prog and [11] pred	All R	115–363	10–50 mg/L/10 mg/L*	[11,102,103,115,116,123]	
CAR	High	HR: N.S.-2.6	HR: N.S.-2.1	All prog	All R	100–201	0.58 and 7.5, 0.58 *	[102,104,124]	
mGPS	High	HR: N.R.	HR: 2.6	Prog	R	132	1	[103]	
Fibrinogen	High	HR: 2.1	HR: 1.8	Prog and pred	R	176	750 mg/dL	[10]	
Albumin	Low	HR: N.R.-1.5	HR: N.S.-1.8	All prog	All R	97–278	35–40 g/L, 35 g/L *	[102,103,114,125]	
C4d	High	HR: 7.3 high vs. low	HR: 0.3 low vs. high	Prog	R	30	1.5 µg/mL	[8]	
Activin A	High	HR: 0.4	HR: 0.4	Prog	R	119	574.0 pg/mL	[55]	
sPD-L1	High	HR: N.R.	H.R.: N.S.	Prog	P	40	0.07–1.83 ng/mL measured at 4 timepoints during therapy	[15]	

BC white blood cell, M-CSF macrophage colony stimulating factor, NLR neutrophil-to-lymphocyte ratio, LMR lympho-cyte-to-monocyte ratio, PLR platelet-to-lymphocyte ratio, IL-6 interleukin 6, CRP C-reactive protein, CAP CRP-to-Albumin ratio, mGPS modified Glasgow prognostic score, C4d Circulating complement component 4d, sPD-L1 soluble programmed cell death ligand 1, HR hazard ratio, N.R. not reported, N.S. not significant, Prog prognostic bi-omarker, Pred predictive biomarker, R retrospective, P prospective, * most frequently used cut-off value.

## Data Availability

No new data were created or analyzed in this study. Data sharing is not applicable to this article.

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
