# Peer review of "Biomarkers for Malignant Pleural Mesothelioma—A Novel View on Inflammation"

_cancers, 2021, doi:10.3390/cancers13040658_

Round 1
Reviewer 1 Report
This review work by Vogl, et al. is highly comprehensive, well-conceived, and well-executed. This reviewer is satisfied with the significance of this review paper. However, the work can be further improved with some concerns to be addressed.
- Line 58-59, please rephrase this sentence to make it easier to understand.
- Line 80, the comma following “proved” should be removed.
- Line 275-289. Recent studies on MPM or malignant peritoneal mesothelioma (very similar to MPM) demonstrated the heterogeneity of PD-L1 level (recommended references: PMID: 33100328, PMID: 33339893) and the factors (e.g. genetic mutations in LATS1/2) regulating its expression (PMID:28344893, PMID: 32824422, PMID: 33240401). It would be better to add the relevant literature to this review paper.
- The authors mentioned the importance of biomarkers-directed therapies in MPM. Could the authors also add some discussions on the potentially responsive biomarkers for immunotherapy?
Reviewer 2 Report
The review titled "Biomarkers for Malignant Pleural Mesothelioma – a Novel View on Inflammation" is a very well-written and thorough review on inflammatory parameters that might be of biomarker value.
I found the review thoroughly enjoyable and educational, a valuable addition to the big list of mesothelioma reviews.
Reviewer 3 Report
This paper is an accurate review on inflammation-derived biomarkers of malignant mesothelioma. In particular, the Authors focus their study on systemic and local inflammation. After an overview of recent and classic studies, they suggest that the suppression of the specific immune system and the activation of its innate counterpart are important drivers of the tumor aggressiveness and mark a poor outcome.
The review length is in accordance with the journal requirements and includes tables/figures as recommended.
The review methodology should be reported. Information about the inclusion and exclusion criteria for the literature search, for example the keywords, the period in which articles have been published, the resulting number of articles, and the final number of articles used for this review should be stated to elucidate the study selection.
The Authors’s conclusions are not novel. Trials using anti-PD1 immunotherapy (e.g. NCT03654833 NCT03918252 NCT0270766) are ongoing, so nothing really new is reported.
Thus, the Authors should delete the sentence "This novel view on the immune system in MPM might help in treating this therapy refractory disease".
Reviewer 4 Report
Malignant pleural mesothelioma (MPM) is closely linked with asbestos exposure and chronic inflammation as one of the major risk factors. The authors have tried to provide a comprehensive review of the inflammatory system, local associated inflammation and systemic inflammatory markers,. The followings points are provided for potentially improving the clinical utility of this review for readers in the clinical settings: Cut-off points of biomarkers mentioned in the manuscript are not very clear for gaining insight for clinical application. Could the authors elucidate the clinical implications of high or low? Definition of biomarkers for clinical implication is important. From clinical perspective, could the review provide some candidates for Human biomarkers? Based on some limited literature searching listed below, serum ferritin seems to be a potential biomarker for malignant pleural mesothelioma? (1) Toyokuni S. Iron addiction with ferroptosis-resistance in asbestos-induced mesothelial carcinogenesis: Toward the era of mesothelioma prevention. Free Radic Biol Med. 2019 Mar;133:206-215. doi: 10.1016/j.freeradbiomed.2018.10.401. Epub 2018 Oct 10. PMID: 30312759. (2) Ohara Y, Chew SH, Shibata T, Okazaki Y, Yamashita K, Toyokuni S. Phlebotomy as a preventive measure for crocidolite-induced mesothelioma in male rats. Cancer Sci. 2018 Feb;109(2):330-339. doi: 10.1111/cas.13460. Epub 2018 Jan 4. PMID: 29193587; PMCID: PMC5797813. (3) Sezgi C, Taylan M, Sen HS, Evliyaoğlu O, Kaya H, Abakay O, Abakay A, Tanrıkulu AC, Senyiğit A. Oxidative status and acute phase reactants in patients with environmental asbestos exposure and mesothelioma. ScientificWorldJournal. 2014 Jan 23;2014:902748. doi: 10.1155/2014/902748. PMID: 24592197; PMCID: PMC3921948. (4) Jiang L, Akatsuka S, Nagai H, Chew SH, Ohara H, Okazaki Y, Yamashita Y, Yoshikawa Y, Yasui H, Ikuta K, Sasaki K, Kohgo Y, Hirano S, Shinohara Y, Kohyama N, Takahashi T, Toyokuni S. Iron overload signature in chrysotile-induced malignant mesothelioma. J Pathol. 2012 Nov;228(3):366-77. doi: 10.1002/path.4075. Epub 2012 Aug 2. PMID: 22864872. Minor points 1. typo-- figure 1 is "Cd4" or "C4d" 2. missing data in Table 2.
Round 2
Reviewer 4 Report
It is suggestive that authors can include ferritin in Table 2 for a comprehensive list of systemic inflammatory biomarkers.